# Investigation into Red Emission and Its Applications: Solvatochromic N-Doped Red Emissive Carbon Dots with Solvent Polarity Sensing and Solid-State Fluorescent Nanocomposite Thin Films

**DOI:** 10.3390/molecules28041755

**Published:** 2023-02-12

**Authors:** Justin B. Domena, Ermin Celebic, Braulio C. L. B. Ferreira, Yiqun Zhou, Wei Zhang, Jiuyan Chen, M. Bartoli, A. Tagliaferro, Qiaxian Johnson, Bhanu P. S. Chauhan, Victor Paulino, Jean-Hubert Olivier, Roger M. Leblanc

**Affiliations:** 1Department of Chemistry, University of Miami, Coral Gables, FL 33146, USA; 2Department of Applied Science and Technology, Politecnico di Torino, 10129 Torino, Italy; 3Department of Chemistry, William Paterson University of New Jersey, 300 Pompton Rd, Wayne, NJ 07470, USA

**Keywords:** carbon dots, red emission, solvatochromic, nanocomposite, thin film

## Abstract

In this work, a NIR emitting dye, *p*-toluenesulfonate (IR-813) was explored as a model precursor to develop red emissive carbon dots (813-CD) with solvatochromic behavior with a red-shift observed with increasing solvent polarity. The 813-CDs produced had emission peaks at 610 and 698 nm, respectively, in water with blue shifts of emission as solvent polarity decreased. Subsequently, 813-CD was synthesized with increasing nitrogen content with polyethyleneimine (PEI) to elucidate the change in band gap energy. With increased nitrogen content, the CDs produced emissions as far as 776 nm. Additionally, a CD nanocomposite polyvinylpyrrolidone (PVP) film was synthesized to assess the phenomenon of solid-state fluorescence. Furthermore, the CDs were found to have electrochemical properties to be used as an additive doping agent for PVP film coatings.

## 1. Introduction

Carbon dots (CDs) have attracted increased attention in various biomedical areas such as in vitro/in vivo imaging, photodynamic and photothermal therapy [1,2,3,4,5,6,7]. When compared to typical organic dyes, CDs have the advantage of low cost, high biocompatibility and photostability, and amphiphilic behaviors [8,9,10,11]. Current efforts in developing red-emissive carbon dots as photosensitizers (PS) past 650 nm have been elusive [12]. The importance of this photophysical property is to surpass the requirement of the utilization of light within the red region to NIR-I window ranging between 650–950 nm, respectively for enhanced biological imaging [13,14]. A multitude of studies have sought to elucidate the synthetic methods to reach red emissive CDs, though few reach past the far-red region. Such efforts involve using carbon sources with features such as extended aromatic systems or heteroatoms (sulfur, nitrogen, and phosphorus) [15,16,17,18,19,20,21,22]. Specifically, the presence of heterocycles has been deduced to contribute to an increased degree of surface defects which effectively reduces the HOMO-LUMO energy gaps leading to a bathochromic shift of emission. In this study, we investigated the increased doping of nitrogen of 813-CDs by the introduction of different chain length PEI. To investigate the electronic transitions of the 813 CDs, the optical band gap energies of the respective CDs were elucidated by using the UV-vis spectrum converted to α vs energy plot by using Equation (1) instead of Tauc’s plot.
(1)$$\alpha =\frac{2.303\;A}{l}$$
where *α* is denoted as the molar absorption coefficient at the resultant wavelength and is calculated from Beer-Lambert’s relation according to Equation (1) such that *A* is the absorbance and *l* is the optical pathlength. The simple use of the Tauc approach is incorrect due to its applicability only to crystals, while CDs should be correctly modelled as large molecules [23,24,25]. Studies have suggested that intricate surface modifications are vital to develop red emissive CDs. Additionally, the interest in developing novel CDs from NIR dye sources has increased greatly [26]. This is due to the notion that when synthesizing a CD, the precursor is only partially broken down during the reaction, leaving moieties close to that of the original base structure. Hence, most of the conjugated network is retained resulting in a slight hypsochromic shift. As promising candidates, NIR fluorophores as precursors to red emissive CDs are of interest. 

Until recently, post-synthetic parameters have been investigated to understand the influence of solvent polarity on the emissive properties of CDs [27,28,29,30,31,32,33,34]. Solvents of various polarities have discreet effects on the electronic transition of dipolar solutes. For example, when exposed to highly polar solvent, a fluorophore can exhibit two types of solvatochromism (positive or negative) [35,36]. Positive solvatochromism occurs due to a lower shift in the UV-vis transition energy resulting in a longer wavelength. This phenomenon is observed because the dipole moment of the molecule in ground state is smaller than that of the dipole moment in the excited state [37]. On the other hand, fluorophores exhibit negative solvatochromism due to a higher shift of UV-vis transition energy resulting in a shorter wavelength [38]. In this work, we propose that this feature of solvent polarity is important as CDs can be exploited in two ways: to produce a red shift in its optical properties, and to be used as a polarity sensing agent. Herein, we report the development of stable red emitting 813-CDs with polarity sensing capabilities.

Polyvinylpyrrolidone (PVP) has extensively been studied for its application in pharmaceuticals, electronics, cosmetics, and polymers [39,40,41,42,43,44,45]. Specifically, as a thin film composite, PVP, when polymerized with CDs, may have the feature to lock in place their emissive state [46,47,48,49], resulting in what is known as solid state fluorescence. There are few papers which address this phenomenon as carbon dots typically undergo fluorescence self-quenching in solid state due to aggregation [50,51]. Furthermore, these 813-CDs have been used with PVP to develop a red emitting thin film nanocomposite polymer which exhibits solid state fluorescence. We hypothesize that by introducing the CDs into a polymer matrix, the disabling of both the π−π interaction and fluorescence resonance energy transfer (FRET) between neighboring CDs will be observed. These interactions are known to play a key role in fluorescence self-quenching of CDs in solid state. Another promising feature of CDs is their potential as a doping agent to enhance electrochemical behaviors in PVP films, namely coatings for battery cathodes.

Herein, we propose the synthesis of far-red emitting carbon dots from the NIR emitting dye, IR-813. In comparison to current NIR photosensitizers, CDs boast features such as thermal and photostability. The feature of these carbon dots stems from the passivation of PEI during synthesis which aids in two aspects: (1) to have increased water solubility, as carbon dots derived from dyes usually lack this characteristic; and (2) PEI as a polymer encases the dye during microwave irradiation to slow the thermal degradation of the dye [52]. A feature of CDs to be used biologically is dependent on their surface functionalization. Specifically, the surface of the CD can modulate the pathway of cellular uptake, intracellular trafficking, and cytotoxicity [53]. As a surface passivant, PEI is a highly favored cationic macromolecule for use in gene transfer therapy due to high transfection efficiency [54]. Additionally, PEI molecules interact with negatively charged proteins in cytoskeletons, such as actin and beta-tubulin [55]. Another important feature of selecting PEI for surface functionalization of CDs is to provide both a positive charge and to improve permeabilization of the plasma membrane. The literature further conjectures that PEI as a drug delivery vesicle or passivant can disrupt endosomal membranes through the well-known “proton sponge effect”, which states that the presence of a weekly basic molecule may cause the endosome to burst. This facilitates the possible pathways that allow binding of the material to DNA [56,57]. To further clarify our rationale for PEI as a choice for nitrogen content, the literature shows that the increase of PEI weight during synthesis serves as a facile method to increase the degree of amidation of a nanoparticle [58].

## 2. Experimental Section

### 2.1. Materials

Two polyethyleneimine (PEI), branched (MW: 600 and 25 K) and *p*-toluenesulfonate were procured from Sigma-Aldrich (St. Louis, MO, USA). Polyvinylpyrrolidone (PVP) was obtained from VWR (West Chester, PA, USA). All solvents used were HPLC grade including ethanol, acetic acid, dimethyl sulfoxide (DMSO), dimethylformamide (DMF), methylene chloride, chloroform, ethyl acetate, toluene, ether, and hexane which were purchased from Sigma-Aldrich (St. Louis, MO, USA), and were used as received without further purification. Deionized (DI) water used was ultrapure (type I) water which was purified using a Millipore Direct-Q 3 water purification system acquired from EMD Millipore Corporation. The purified water displayed a surface tension of 72.6 mN m^−1^, a resistivity of 18.2 MW cm and a pH value of 7.0 ± 0.3 at 20.0 ± 0.5 °C. All the chemicals were used as received. 

### 2.2. Synthesis of 813-CD-600 PEI

The 813-CDs were obtained by means of microwave pyrolysis of IR813 (0.100 g) and PEI MW 600 (0.050 g) as illustrated in Appendix A (in the Appendix A). To preface, the precursors were dispersed in a beaker containing 15 mL of methanol and sonicated for 1 min to ensure a homogenous mixture which was observed to exhibit a dark-green color. The solution of starting material was then placed in the microwave and set for 360 s at a power setting of 700 W. Post pyrolysis, it was observed that no solution remained, leaving a bright orange gel-like product, 813-CD (Appendix A). The product was then purified by flash chromatography with an eluent system of 75% hexane, 5% methanol, and 20% ethyl acetate by volume (v:v:v), producing an orange-red product labelled 813-CD-600 PEI. To ensure a product with little to no solvent, a Buchi Rotavapor (Waterbath B-480) was used to eliminate all organic solvent impurities. The purified solution of 813-CDs was dried and re-dispersed in ultrapure (type I) deionized (DI) water. The dispersion was then frozen at −40 °C for 24 h and sequentially set for lyophilization for 72 h. Producing again, an orange-red gel-like product. These freeze-dried CDs were then used for study.

### 2.3. Synthesis of 813-CD-25K PEI

The 813-CDs were obtained by means of microwave pyrolysis of IR813 (0.100 g) and PEI MW 25K (0.050 g). The methodology for the synthesis of 813-CD-25K is the same as in Section 2.2. Post pyrolysis, it was observed that no solution remained, leaving a dark black-orange 813-CD (Appendix A). The product was then purified by flash chromatography with the same eluent system as 813-CD-600 PEI, producing a dark green-orange product labeled 813-CD-25K PEI. These freeze-dried 813-CD-25K PEI CDs were also used in our study.

### 2.4. Synthesis of 813-CD-600 PEI/PVP Nanocomposite Thin Film

To synthesis 813-CD-600 PEI/PVP nanocomposite thin films, 500 mg of PVP were dissolved in 5 mL of ethanol and sonicated for 1 min. Next, 5.00 mg of 813-CD were also dispersed in 5 mL of ethanol and sonicated for 1 min. The now dispersed solution of 813-CDs was aliquoted into the prepared PVP ethanol solution at 1 mL intervals and was uniformly dispersed under stirring. The obtained mixture was poured into a glass Petri plate and dried for 24 h under ambient conditions in the dark to obtain the 813-CD/PVP film.

### 2.5. Characterization

UV-vis spectra were obtained from an Agilent Cary 100 UV-vis spectrophotometer. Photoluminescence (PL) characterization was performed on a Fluorolog HORIBA Jobin Yvon fluorometer with a slit width of 5 nm for excitation and emission. All optical characterization spectra were obtained with quartz cells possessing an optical pathlength of 1 cm. Fourier-transform infrared (FTIR) spectroscopy data were obtained with a PerkinElmer FTIR (Frontier) spectrometer (Waltham, MA, USA) by using the attenuated total reflection (ATR) technique with air as background. The AFM images of 813-CDs were obtained with an Agilent 5420 atomic force microscope (Santa Clara, CA, USA) by using tapping mode. To perform AFM measurements, a drop of diluted 813-CDs aqueous solution was applied on a clean silica mica slide and air dried, which was then transferred to perform the screening using tapping mode. The tip used was silicon (length: 225 μm; thickness: 5 μm) manufactured from Nanosensors with a force constant of 15 N/m. The TEM was performed by using a JEOL 1200 EX TEM (Peabody, MA, USA). For TEM measurements, a drop of the 813-CDs solution was placed on a carbon coated copper grid and air-dried prior to examination. Scanning electron microscopy (SEM) images were taken at 15 KV accelerating voltage on a Hitachi SU1510. The Zeta potential was recorded on a Malvern Zetasizer nano-series. Electrochemical measurements were conducted using a PARSTAT 3000A potentiostat (Ametek Scientific Instruments, Berwyn, PA, USA) as reported in the literature [59].

## 3. Results and Discussion

### 3.1. UV-Vis Analysis of 813-CD 600 PEI

The detailed preparation of 813-CDs was described in Section 2. The UV-vis spectroscopy is a suitable tool to assess the electronic structure of conjugated systems. As expected, the absorption spectra of 813-CDs exhibited the typical absorption bands attributed to the π−π * transitions of the C=C bond and the n−π * transition of the C=O bond at 210 and 290 nm, respectively (Figure 1) [60,61]. A band at 375 was found to be similar to that of IR-813, which derives from the n−π* transition of C=N functionalities present in the cyclic systems. Additionally, further bands were found at 450, 545, 590, 665, and 745 nm, respectively. These ranges are hypothesized to be due to the degradation of the cyclic structure of IR-813 to form new nitrogen-containing heterocyclic structures coordinated with PEI to form new emissive states [62,63,64,65]. However, the loss of conjugation is what leads to bands with a hypsochromic shift. 

### 3.2. Solvatochromic PL Behavior of 813-CD

Fluorescence spectroscopy is both an ideal and sensitive technique to supplement data from UV-vis to understand the fluorescent behavior of materials that absorb light. The farthest PL peak and its respective PL intensity in counts per second (CPS) of 813-CD-600 PEI (0.1 mg/mL) was recorded to assess the viability of the material in solvent polarity sensing techniques across a wide range of excitation in decreasing solvent polarities in Table 1 (water, ethanol, acetic acid, DMSO, DMF, methylene chloride, chloroform, ethyl acetate, toluene, ether, and hexane, respectively). The expected result was to produce CDs that retained optical properties similar to IR-813, which has notable emission peaks at 600 and 827 nm, respectively. We hypothesized that there may be degradation of the dye during the synthetic process, resulting in slight loss of conjugation. It is this decrease in conjugation that contributes to an increase of the energy band gap, producing a hypsochromic shift. Interestingly, this behavior was observed when the PL of the 813-CDs were excited between 380 to 600 nm, at 20 nm intervals in comparison to IR-813 (Appendix A). To our surprise, the most polar solvent (water) 813-CDs were found to possess far-red emission at 698 with a PL intensity of $1.5\times 10^{6}$ (CPS). Interestingly, 813-CDs were found to have a clear isosbestic point at 575 nm. This finding is important because it means that there will be no change in the optical property at a set wavelength and hence, a uniformity in optical properties across the size distribution of CDs. This agrees with our prior hypothesis regarding the emissive origin of our CDs. As the 813-CDs were introduced to environments of decreasing solvent polarity, we started to observe negative solvatochromism. In ethanol, a hypsochromic shift was immediately observed as the farthest emissive peak was found at 650 nm with a PL intensity of $1.4\times 10^{6}$ (CPS). The same trend occured as the 813-CDs were exposed to environments with even lower solvent polarity. In acetic acid, the PL shifted to 655 nm with a slight increase of PL intensity ($1.9\times 10^{6}\;\left(\mathrm{CPS}\right)$). The DMSO possessed PL at 610 nm and had a quite drastic decrease of PL intensity of $5.0\times 10^{5}\;$(CPS). Serving as the midway point of decreasing polarity, both DMF and methylene chloride produced the familiar PL peak at 655, but had a noticeable decrease of PL intensity to $1.1\times 10^{6}\;\mathrm{and}\;1.0\times 10^{6}$ (CPS), respectively. In chloroform, the 813-CDs emitted at 653 nm, but drastically dropped in PL intensity to $2.5\times 10^{5}$ (CPS). Following the same trend, ethyl acetate, ether, and toluene had observed PL shifts to 650 nm followed by low PL intensities of $8.0\times 10^{5},\;7.2\times 10^{5}$, and $3.8\times 10^{5}$ (CPS), respectively. Notably, in hexane (the lowest polarity), the farthest emissive peak was found to be at 590 nm, meanwhile having the low PL intensity of $4.5\times 10^{5}$ (CPS).

To address the decrease in PL intensity with decreasing solvent polarity, we hypothesized that in more polar solvents there is the presence of aggregation induced emission (AIE) by particles. This phenomenon leads to more intense emissions at aggregated states in comparison to conventional aggregation fluorescence quenching mechanisms displayed by CDs [66]. From the results of the fluorescence study in various solvents, we propose a schematic illustration of the emission mechanism from 813-CDs in solvents of decreasing polarity (Figure 2). We hypothesized that behavior of the excited-state of 813-CDs is dictated by the solute-solvent interactions of various polarities. Specifically, under excitation the solvent molecules have the opportunity to rotationally reorient to stabilize the excited-state dipole moment. This phenomenon as depicted is known as solvent relaxation, which is a key feature of both positive and negative solvatochromism [67,68,69]. Under this condition, we believe that 813-CDs experience a lowering of the energy of their excited state. Post-emission, the excited 813-CD returns to ground state. According to the literature, as the emission of CDs are linked to their $\pi \to \pi {\;}^{*}$ transition, the polarity of the solvent used will result in a reduction of energy of the first excited state (S_1_) than that of the ground state (S_0_). Effectively, as solvent polarity increases, the energy gaps between S_0_ and S_1_ are reduced resulting in longer emitted wavelengths. This phenomenon is attributed to the behavior of more polar solvents to enhance the electron cloud density of the CD’s sp^2^ domain which modulates the $\pi \to \pi {\;}^{*}$ electron transition [70]. We also observed the presence of the double-peaked nature of the 813-CDs. We hypothesized that this characteristic is due to a mix of the carbon core state (600 nm) and surface defects from PEI on the surface (697 nm). This is attributed to the transition of a higher singlet electronic state (S_2_) undergoing internal conversion to a lower singlet electron state S_1_ before returning to S_0_. As we observed the drastic red-shifts of 813-CD-25K PEI, the idea of emissive states due to surface defects made sense as the double-band peak was not consistent in comparison to 813-CD-600 PEI having red shifts as far as 776 nm [71].

### 3.3. Optical Manipulation of 813CDs: 600 PEI vs. 25K PEI

To understand the influence of nitrogen content on the optical properties of the 813-CDs, α vs energy plot was developed as previously described, to reveal favorable insight into the electronic transitions between that of 813-CD 25K PEI and 813-CD 600 PEI via UV-vis spectra. The UV-vis data used in the α vs energy plot for determining the HOMO-LUMO energy gap ($\Delta_{\mathrm{HL}}$) are provided in Appendix A. An analysis of 813-CD 25K PEI revealed that there were four main electronic transitions between the HOMO-LUMO states of the CD at Eg values of 5.02, 4.29, 2.35, and 2.26 eV, respectively (Figure 3A,B). We propose that these electronic transitions are due to the type of doped group, such as intrinsic carbon, graphitic nitrogen, carbonyl, and increased surface amines present, respectively. Our premise is that the increased amount of surface defects from the orientation of nitrogen-based cyclic structures and amines on the surface of the CDs results in the lowering of the energy band gap; thus, causing the emission to be red shifted as the stokes shift range changes. To validate our hypothesis, 813-CD 600 PEI was then screened as the material of choice for α vs energy plot for determination as there was less nitrogen present in the same reaction conditions. Therefore, we expected that the band gaps overall would increase slightly. As expected, the analysis of 813-CD 600 PEI revealed that there were four electronic transitions between the HOMO-LUMO states of the CD that resembled that of the prior α vs energy plot. Only the $\Delta_{\mathrm{HL}}$ was slightly larger, providing Eg values of 5.26, 4.34, 4.01, and 2.75 eV, respectively (Figure 3C,D). This observation made sense as even though there were the same type of doped groups present (intrinsic carbon, graphitic nitrogen, carbonyl, and surface amines present, respectively), with the lowering of nitrogen content, there were fewer surface defects state present which would lead to a higher energy band gap.

Subsequently, the PL properties between both 813-CD 25K PEI and 813-CD 600 PEI in water were studied to reveal how the $\Delta_{\mathrm{HL}}$ correlates to the fluorescence capabilities. An observation of the PL spectrum of 813-CD 25K PEI postulates that the lowered band gap as determined by the α vs energy plot leads to the far-red emission at 723 nm (Figure 4A). A second emissive peak was also observed at 600 nm, which decreased in intensity as the excitation wavelength was swept from 400 to 780 nm in 20 nm intervals, respectively. Interestingly, 813-CD 25K PEI was found to exhibit excitation dependence which is common with CDs that contain increased surface defects. Additionally, the normalized PL spectrum revealed that as the excitation wavelength is increased, the far-red emissive peak is further red shifted as far as 776 nm (Figure 4B). As expected, the PL spectrum of 813-CD 600 PEI was quite different regarding its far-red peak which was observed at 697 nm (Figure 4C). This result was expected in correlation to the α vs energy plot which suggests that the $\Delta_{\mathrm{HL}}$ is at a higher value. At lower excitation wavelengths, 813-CD 600 PEI exhibited excitation dependence with rising emissive peaks at 600, 557, 510, and 445 nm, respectively (Figure 4D). We posited that during synthesis, the smaller chained PEI formed a variety of heterocyclic structures at the surface of the CD which led to the PL behavior as described. 

There are quite a few perspectives on what factors serve as the emissive origin of CDs. All reasons point towards factors that are outside the realm of quantum confinement that associate the emissive shifts of well-defined nanoparticles according to size. The structure-property relationship of carbon dots is vague, leading to challenges in elucidating the source of emission. In fact, there are three models currently debated as plausible mechanisms for emission in CDs, i.e., carbon core states, surface states, and molecular fluorescence. Neither are well defined due to the diversity (size distribution) of CDs which cannot be isolated into just one uniform product. The CDs are a heterogenous distribution of different-sized particles that have the same emissive peak, loosely displaying the quantum size effect. Our line of thought stems from a mix of both carbon core and surface states of carbon dots. We posited that 813-CD-600 PEI may have more sp^2^ domain sizes (carbon core state behavior) which correlates to the slight size increase, though does not particularly contribute towards red emission [72]. This idea is corroborated by 813-CD-25k PEI which boasts farther red emission, yet has a smaller, more uniform size distribution. Hence, we hypothesized that the CD’s emission may be linked to surface defects states of the CDs due to the protection of PEI of IR-813 [73].

The FTIR serves as a powerful tool to qualitatively analyze both the core and the functional groups that passivate the surface of CDs, which typically contain oxygen and nitrogen functional groups. A comparative analysis of 813-CDs and the starting material was performed to reveal insight into the surface of the CD (Figure 5). At first glance, both 813-CD 600 and 25K PEI have similar peaks. Upon analysis of 813-CDs, the intense peaks at 1335–1372 cm^−1^ and 1030–1070 cm^−1^ are due to the asymmetric S=O and sulfoxide stretching vibrations, respectively. This characteristic makes sense as there would be sulfoxide moieties present from the IR-813 precursor, which have the same respective peaks. These results suggest that 813-CDs retain most functionality from IR-813. A range between 3000–3100 cm^−1^ is attributed to the C-H stretching of the sp^2^ carbon species, which make up the carbonized network of 813-CDs. Additionally, the broad peak at 3480 cm^−1^ is attributed to stretching vibrations of O-H (trace water) present, as well as the medium peak at 2850–3000 cm^−1^ indicative of C-H stretching of the alkane groups due to PEI present during the synthetic process, confirming the availability of these groups on the surface of the 813-CDs. The peak at 3300 cm^−1^ is ascribed to N-H (secondary) group on the surface of the 813-CDs due to PEI. It is hypothesized that these groups, which are abundant on the surface, are responsible for the high solubility of 813-CDs in water. This finding was reinforced through the analysis of 813-CD 25K PEI that had a more apparent N-H peak at 3300 cm^−1^, attributed to the increase of nitrogen from the larger chain size, as well as more trace water found at 3480 cm^−1^.

Functional moieties of two types of CDs can be identified and quantified by TGA and DTG analyses by their unique decomposition temperatures in reference to those of precursors. In general, the TGA of CDs and PEI (Figure 6A) showed similar thermal decomposition trends across different temperatures, which demonstrates a structural similarity among CDs and PEI. In comparison, the TGA of IR-813 exhibited more variations between 40–250 °C, which corresponds to a complicated structure. In addition, the differences of CDs and their precursors are displayed by the DTG (Figure 6B) and show that: (1) IR-813 has six decomposition stages: 40–127, 127–200, 200–248, 248–376, 376–463, and 463–673 °C; (2) PEI 600 has two decomposition steps: 40–463, and 463–673 °C; (3) PEI 25K has two decomposition steps: 40–376, and 376–673 °C; (4) the CDs (CD-PEI 600) possess four decomposition periods: 40–127, 127–248, 248–463, and 463–673 °C; (5) the CDs (CD-PEI 25K) possess four decomposition periods: 40–200, 200–376, 376–463, and 463–673 °C; (6) both the DTG of PEG 600 and 25K generally show two decomposition stages, but different ranges indicate the effect of molecular weight on the thermostability of polymers; and (7) with the DTG of a well-established CD species as reference, any peaks between 40–127 °C indicate the loss of water molecules adsorbed on CDs. The mass loss at 127–200 °C suggests the loss of water molecules generated through intramolecular dehydration condensation reactions. The stage between 200–248 °C is attributed to the decomposition of edge-plane oxygen-containing functional groups. The stage at 248–376 °C is due to the decomposition of relatively stable oxygen-containing functional groups, and sublimation of small carbon frameworks. The stage at 376–463 °C indicates the decomposition of amines. Eventually, the stage at 463–673 °C indicates the decomposition of graphene-like structures; (8) therefore, during 40–127 °C, IR-813 and CDs (CD-PEI 600) showed desorption of moisture by 6 and 9%, respectively. In addition, in the corresponding TGA (Figure 6A), mass loss was observed among all the other samples, which was expected due to the presence of amines; (9) during 127–200 °C, CD-PEI 600 and CD-PEI 25k exhibited 13% and 12% mass loss of water molecules, respectively, formed through intramolecular dehydration condensation reactions. However, since intramolecular dehydration condensation reactions cannot occur in PEI, during this period, both PEI did not display significant mass loss. Furthermore, IR-813 contains perchlorate anions, so the mass loss is hypothesized to be evaporation of perchloric acid considering its boiling point at 203 °C; (10) the peak shown in the DTG of IR-813 between 200–248 °C indicates the loss of oxygen-containing functional moieties formed during the process of dye decomposition (Figure 6B); (11) sharp decompositions of relatively stable oxygen-containing functional groups and carbon frameworks were observed in all the samples during 248–376 °C, and the corresponding mass percentages in IR-813, PEI 600, PEI 25K, CD-PEI 600, and CD-PEI 25K are 40%, 59%, 85%, 30%, and 61%, respectively; (12) subsequently, mass losses occurred during 376–463 °C due to the decomposition of amines and the mass losses were 11%, 8%, 3%, 11%, and 5% for IR-813, PEI 600, PEI 25K, CD-PEI 600, and CD-PEI 25K, respectively; (13) and in the last stage between 463–673 °C and above 673 °C, mass losses of 30%, 18%, 7%, 27%, and 13% were obtained for IR-813, PEI 600, PEI 25K, CD-PEI 600, and CD-PEI 25K, respectively, due to the decomposition of aromatic structures. 

Preliminary consideration of the structure should be based on both TGA and FT-IR data. As shown by TGA data, the aromatics domains represent a relevant amount of both 813-CDs produced by using both PEI 600 and 25 K, and they are due to native aromatic moieties of carbon dots and piperazine-like structure formed during the degradation of PEI [74]. Additionally, IR analysis shows the signal of sulphate proving the persistence of tosyl residues inside the 813-CDs.

Accordingly, we assumed the simultaneous occurrence of three pathways regarding the evolution of PEI, IR-813 and tosyl residues during the production of 813-CDs, as reported in Figure 7. 

The PEI degradation is well described in the literature, as reported by Kosheleva et al. [74]. with two possible degradative mechanisms under thermal stimuli. The heterolytic cleavage of PEI promotes the formation of terminal alkenes (**2**) and primary amines (**3**) while the intramolecular cyclization promotes the production of piperazine derivatives (**5–6**). The IR-813 evolution is more complex, and it is based on cycloaddition, nucleophilic additions, and the aromatization process. Firstly, we observed that the pathway that converts **7** into **9** through cycloaddition is not favorable due to the spiro [5.5] undecane moieties (**9**) that are a great stress point for the whole structure. 

The equilibrium from 7 to **10** forms cis-diene that could condense producing **12** or reaching the equilibrium with **13**. Both **12** and **13** contain a tertiary carbocation that could be a center for nucleophilic addition of amine groups producing the key intermediate **15**. This species could be rearranged through aromatization forming **18** and **19** that are representative of the ramification of the 813-CDs structure, where the link center based on IR-813 is combined with PEI fragments and other IR-813-based units. Furthermore, protonated tosyl species (**20**) could undergo a heterolytic radical cleavage forming **21** and **22**. Sulfanyl species could promote radical condensation leading to the formation of organic highly oxidated organosulphur compounds (**24** and **25**) properly inserted in the 813-CDs. 

The zeta potential of the 813-CD-600 PEI was obtained to further understand both the surface moiety and the surface charge (Table 2). This technique is vital when elucidating the colloidal stability of CDs in various solvents. To preface, an absolute value above 20 mV typically results in CDs that participate in interparticle repulsion [75], effectively producing a well-dispersed colloidal solution. The 813-CD-600 PEI was found to have a negative potential of −5.11 mV, which suggests that the CDs are composed of negative functional groups such as carboxylic and sulfates on the surface attributed to the functionality left by IR-813 (Appendix A). Hence, the zeta potential data suggests that the CDs will experience the formation of aggregates in solution, meaning an observation of the aggregates by both the TEM and AFM measurements. A zeta potential analysis of 813-CD 25K PEI revealed a value of +18.8 mV, which corresponds to prior data that postulated that this CD had an increased amount of surface amines that contributed to the positive charge of the material (Appendix A). Furthermore, the value of the zeta potential suggested that there would be uniformity among the CDs with very little aggregation, as confirmed by both TEM and AFM measurements.

The TEM images were studied regarding 813-CDs to understand the X−Y plane size distribution. The sample was sonicated for 15 min prior to measurement to break down any aggregate formation. Analysis indicated that the 813-CD 600 PEI showed a narrow size distribution of 3.0–10.0 nm with a mean size of 5.5 nm (Figure 8A). Additionally, the histogram determined a particle count of over 50 particles with an acceptable degree of uniformity (Figure 8B). The AFM was also performed on the 813-CDs to understand the height profile of the CDs in the z-axis (Figure 8C). The AFM images show that the 813-CD 600 PEI particles are 6−6.5 nm in height, consistent with the previous TEM diameter distribution and confirming the quasi-spherical structure of the 813-CDs. To our surprise, the TEM measurements 813-CD 25K PEI revealed a much smaller and more uniformed narrow size distribution of 2–3.5 nm with a mean size of 2.8 nm (Figure 8D). The histogram of over 500 particles indicates a large degree of uniformity (Figure 8E). These results agree with the AFM measurements which indicate a height profile of roughly 1–2.6 nm (Figure 8F). We propose that the stark difference in size of 813-CD 25K PEI is due to the nature of the larger chained polymer. Specifically, the polymer encases the IR-813 dye during the synthetic process and acts as a barrier to discourage the high degree of carbonization resulting in a smaller particle size, as reported in the literature [76]. In contrast, 813-CD 600 PEI, having a smaller chained polymer, offers less protection; therefore, resulting in a higher degree of carbonization that increases the particle size of the respective CDs.

### 3.4. Synthesis of 813-CD/PVP Nanocomposite Film

The detailed preparation of 813-CD/PVP nanocomposite thin films were described in Section 2. To assess the solid-state fluorescence of the CDs locked in the PVP polymer matrix, a 535 nm laser was used as a source of excitation onto the 813-CD/PVP film. When light was placed incident onto the face of the thin film, a qualitative bright-orange fluorescence was observed immediately. This phenomenon can be attributed to the absorbance of light by 813-CDs which then emit photons at a longer wavelength, as proposed in Figure 9A. This is in agreement with the literature, which confirms that typical organic fluorophores locked in polymer matrices can attain high optical efficiency due to the spacing of the molecules, discouraging FRET or self-quenching [77,78,79]. The SEM is a vital tool to elucidate the morphology of the material. As seen in the SEM image (Figure 9B), structural modifications take place in the sample due to interactions of 813-CDs with PVP to produce a surface with minimal defects. The PL of the film produced peaks at both 600 and 650 nm, respectively under 540 nm excitation (Figure 9C).

### 3.5. Cyclic Voltammetry Assessment of 813-CD

The electrochemical behavior of 813-CD as a doping agent for PVP nanocomposites were assessed for applications towards the potential of being used as a coating additive for cathodes to enhance electrochemical performance. In this respect, the literature has shown that PVP film coatings are utilized as additives for a reducing agent to obtain improved stability and rate of lithium ion batteries [80,81]. The presence of these peaks corroborates the n-type nature of our 813-CDs motifs, and are likely a product of the direct reduction of discrete components within the CDs, such as -NH_2_ from PEI which coats the surfaces [82,83,84,85]. The analyzed system displayed two distinct cathodic peaks at −0.11 V (E_p*i*1_) and −0.51 V (E_p*i*2_). These electrochemical signals indicate that the 813-CDs feature two distinct electronic states under electronically reducing conditions (Figure 10A). The respective oxidation peaks were observed at 0.43 V(E_p*i*3_) and −0.45 V (E_p*i*4_) in the reverse scan sweep. Additionally, the stability of 813-CDs was assessed in cycle scans to understand its usability as an additive for PVP coatings for cathodes. To our surprise, the steady decrease of redox peak intensities was not noticeably significant over time (Figure 10B). We suggest that due to the increased stability of our 813-CDs and the strong secondary interactions that exist in the system, no major structural reconfiguration occurs as a function of potential applied, a feature indicative of electrochemical stability [84,85,86]. 

## 4. Conclusions

For the first time, we have synthesized a new type of red-emitting CDs that have been derived from the organic fluorophore IR-813. Our studies have found that the red emission is attributed to the decrease of conjugation of the fluorophore moieties that passivate the CD alongside emissive states derived from PEI. Further, we prove that the 813-CDs can be applicable in solvent polarity sensing, as well as inducing a bathochromic shift in water. A summary of the synthetic protocols and properties for each CD is provided in Appendix A. Additionally, this is the first report of far-red emissive CD/PVP nanocomposite films with solid state emission. To our surprise, the CV assessment of the 813-CDs concludes that this material is suitable as a doping agent to increase the electrochemical behavior of PVP to be potentially applied as a dopant for cathode film coating. In further application, we believe that the results of our far-red emissive carbon dots are highly favorable. The prospects of 813-CDs in vivo are of utmost importance moving forward to assess the cytotoxicity in cell lines. We are hopeful that once validated, our CDs are to be used as theranostic agents to both image and deliver drugs in vivo with live fluorescence tracking. Recent work from our group involving cationic CDs has shown the capabilities of positively charged CDs to bind with DNA both through electrostatic and intercalative modes [87]. We hypothesize in this regard that our CDs may be applicable in this field of study in future works.

## Figures and Tables

**Figure 1 molecules-28-01755-f001:**
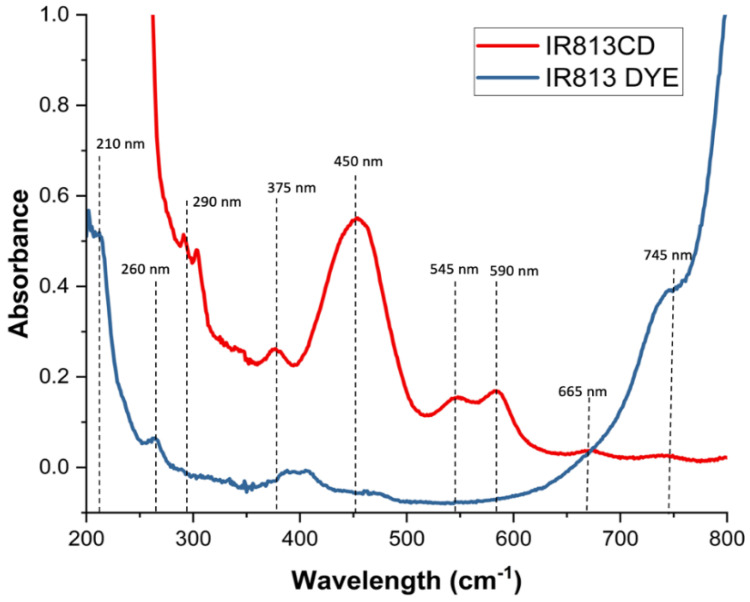
UV-vis analysis of 813-CDs and IR-813 dye.

**Figure 2 molecules-28-01755-f002:**
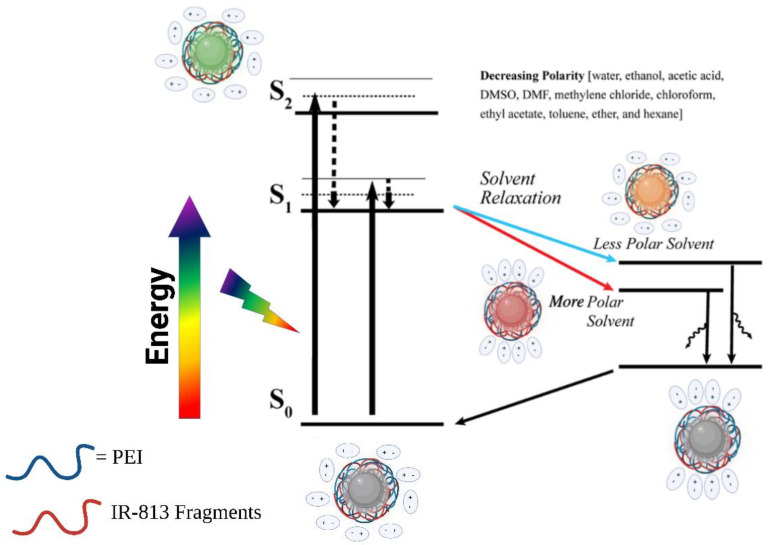
Schematic illustration of the emission mechanism from 813-CDs in solvents of decreasing polarity. Created with BioRender.com, accessed on 1 January 2023.

**Figure 3 molecules-28-01755-f003:**
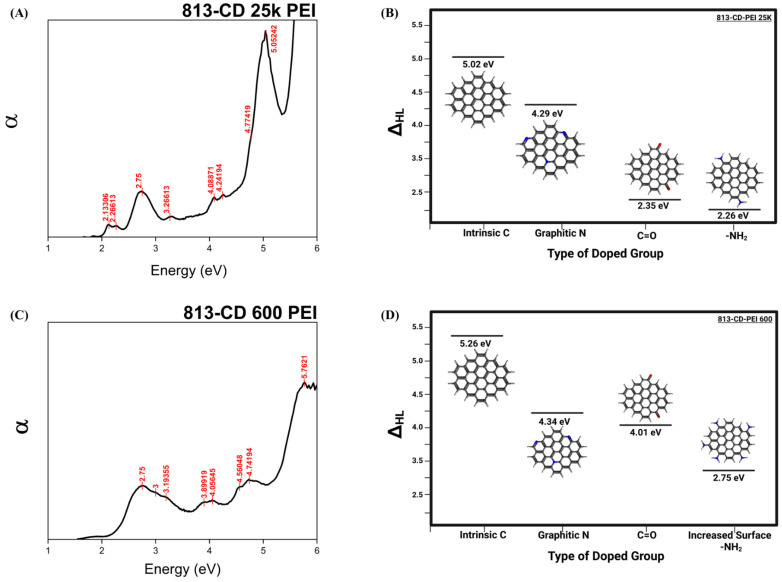
α vs energy plot of (**A**) 813-CD 25k PEI. (**B**) model representation of $\Delta_{\mathrm{HL}}$ versus type of doped group of 813-CD 25k PEI. (**C**) α vs energy plot of 813-CD 600 PEI. (**D**) Model representation of model representation of $\Delta_{\mathrm{HL}}$ versus type of doped group of 813-CD 600 PEI. Created with BioRender.com, accessed on 1 January 2023.

**Figure 4 molecules-28-01755-f004:**
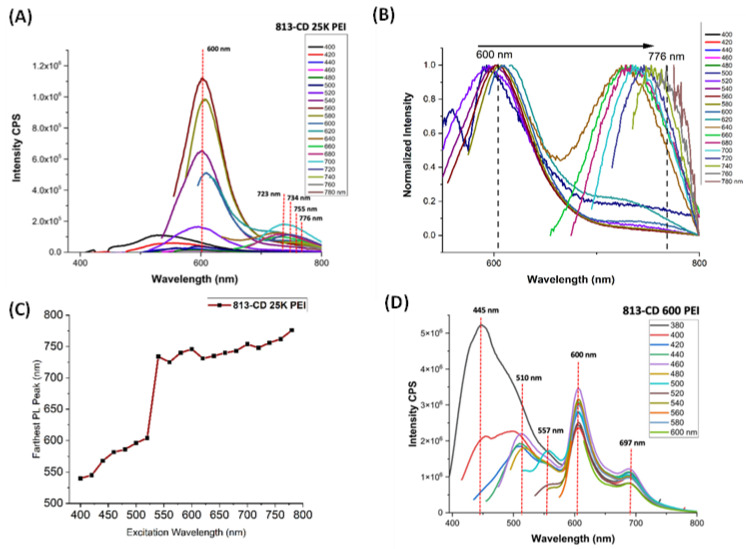
PL spectra of 813-CDs. (**A**) PL spectrum of 813-CD 25k PEI. (**B**) Normalized PL spectrum of 813-CD 25k PEI. (**C**) Plot of the second PL peak for 813-CD-25K PEI vs. excitation wavelength (nm). (**D**) PL spectrum of 813-CD 600 PEI.

**Figure 5 molecules-28-01755-f005:**
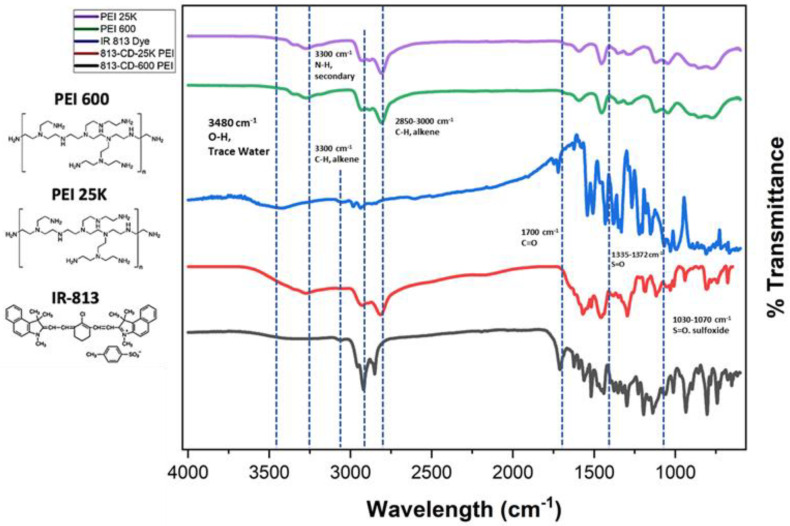
FTIR off-set spectra of PEI 25K, PEI 600, IR-813 dye, 813-CD-25k PEI, and 813-CD-600 PEI, respectively.

**Figure 6 molecules-28-01755-f006:**
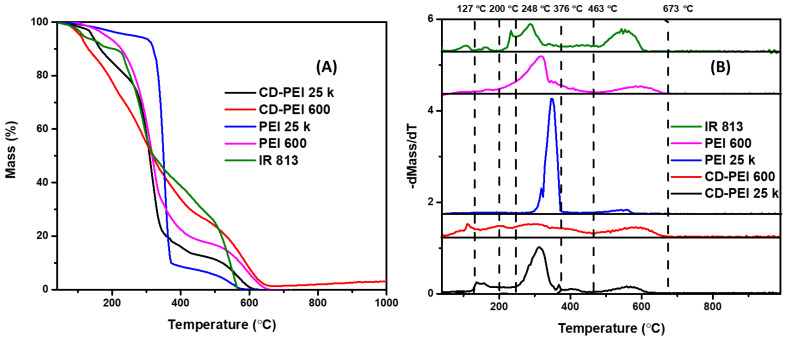
TGA (**A**) and DTG (**B**) of CDs and precursors including IR-813, PEI 600 and 25K.

**Figure 7 molecules-28-01755-f007:**
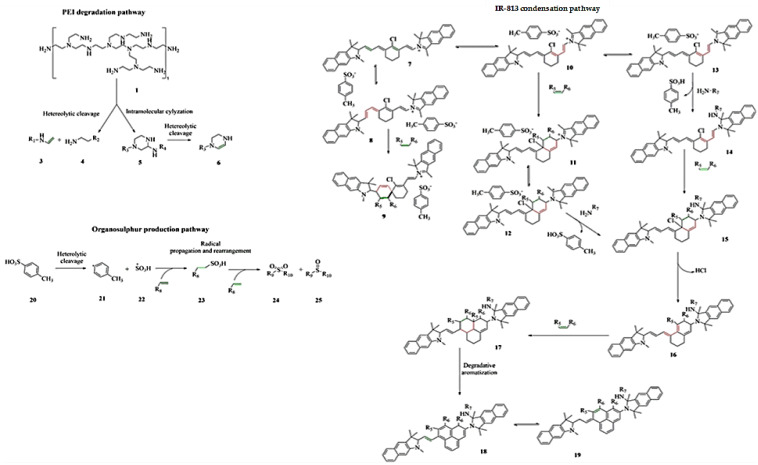
Formation mechanism of 813-CDs.

**Figure 8 molecules-28-01755-f008:**
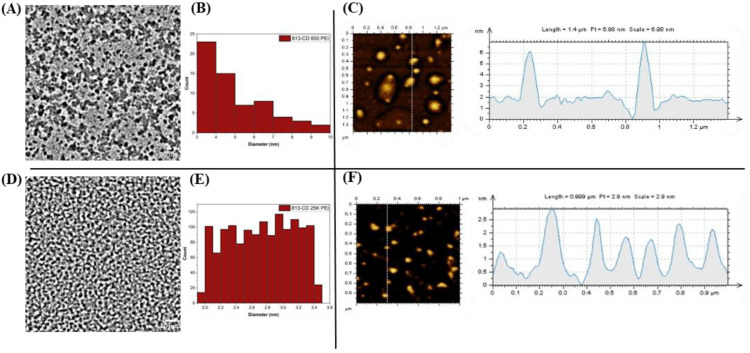
TEM and AFM measurements of 813-CDs. (**A**) TEM of 813-CD-600 PEI. (**B**) AFM scan of 813-CD-600 PEI. (**C**) Histogram of 813-CD-600 PEI. (**D**) TEM of 813-CD-25K PEI. (**E**) AFM measurement of 813-CD-25K PEI. (**F**) Histogram of 813-CD-25K PEI.

**Figure 9 molecules-28-01755-f009:**
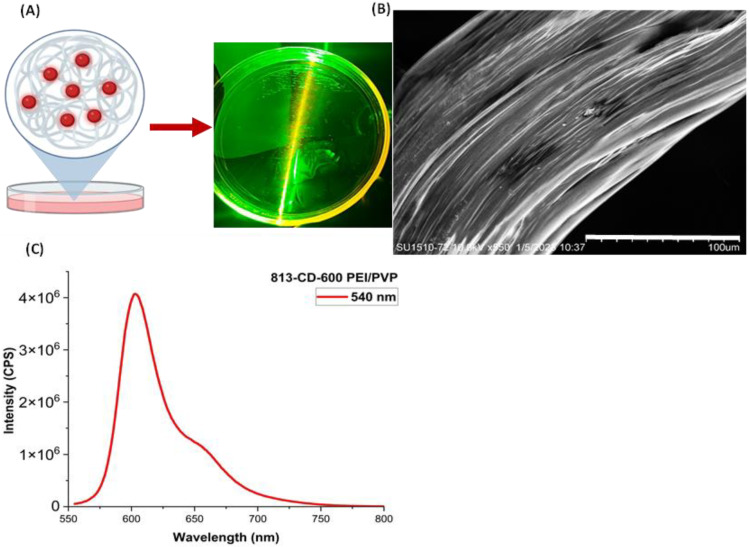
(**A**) Graphical representation and qualitative assessment of 813-CD/PVP nanocomposite thin films on a glass petri dish under 535 nm excitation. Created with BioRender.com, accessed on 1 January 2023. (**B**) SEM measurement of 813-CD/PVP nanocomposite thin film. (**C**) PL spectrum of 813-CD/PVP nanocomposite thin film under 540 nm excitation.

**Figure 10 molecules-28-01755-f010:**
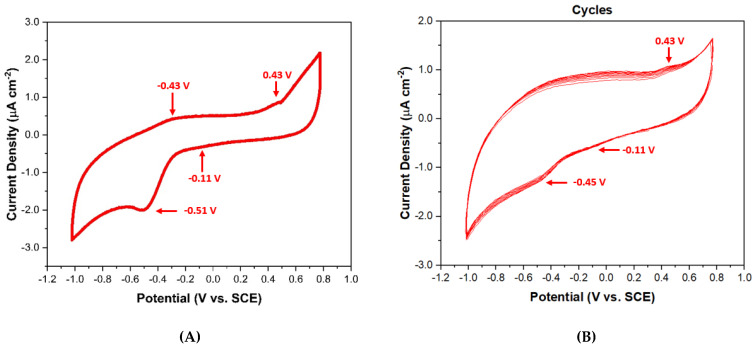
Cyclic voltammograms of 10 µg/mL of 813-CDs (**A**) at 25 mV/s scan rate (**B**) in repeat cycles recorded in H_2_O using 0.1 M NaCl as supporting electrolyte, glassy carbon (GC) electrode as working electrode, Ag/AgCl as reference electrode, and a Pt wire as a counter electrode.

**Table 1 molecules-28-01755-t001:** Optical data of 813-CDs (farthest PL and intensity) and solvents used with their respective polarity index.

OPTICAL ANALYSIS OF 813 CDS IN VARIOUS SOLVENTS OF DECREASING POLARITY			
SOLVENT	Polarity Index (*P)*	Farthest PL (nm)	PL Intensity (CPS)
**WATER**	**1.00**	**698**	** $1.5*10^{6}$ **
**ETHANOL**	**0.654**	**650**	** $1.4*10^{6}$ **
**ACETIC ACID**	**0.648**	**655**	** $1.9*10^{6}$ **
**DMSO**	**0.444**	**610**	** $5.0*10^{5}$ **
**DMF**	**0.386**	**655**	** $1.1*10^{6}$ **
**METHYLENE CHLORIDE**	**0.309**	**655**	** $1.0*10^{6}$ **
**CHLOROFORM**	**0.259**	**653**	** $2.5*10^{5}$ **
**ETHYL ACETATE**	**0.228**	**650**	** $8.0*10^{5}$ **
**ETHER**	**0.117**	**650**	** $7.2*10^{5}$ **
**TOLUENE**	**0.099**	**650**	** $3.8*10^{5}$ **
**HEXANE**	**0.009**	**590**	$4.5*10^{5}$

**Table 2 molecules-28-01755-t002:** Zeta potential of respective 813-CDs.

Carbon Dots	Zeta Potential (mV)
813-CD-600 PEI	−5.11 mV
813-CD-25K PEI	+18.8 mV

## Data Availability

Not applicable.

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
