# Peer review of "Investigation into Red Emission and Its Applications: Solvatochromic N-Doped Red Emissive Carbon Dots with Solvent Polarity Sensing and Solid-State Fluorescent Nanocomposite Thin Films"

_molecules, 2023, doi:10.3390/molecules28041755_

Round 1
Reviewer 1 Report
Leblanc et al. report on the preparation and solvatochromism of carbon dots (CDs) derived from the NIR dyes, p-toluenesulfonate (IR-813); and the nitrogen content can be increased by the introduction of polyethyleneimine (PEI). However, some important issues have to be well addressed. For this reason, I cannot recommend the publication of the manuscript in its present form.
(1) The dyes themselves have excellent optical properties. The authors are suggested to explain the advantages of dye-derived CDs. What’s more, I didn’t see the improved optical properties of the 813-CDs in this manuscript with respect to their precursors. Probably, the authors can compare their CDs with IR-813 in the case of quantum yields (QYs), stability, etc.
(2) For the evaluation of energy gap by Tauc Plots, it will be better to indicate some references, e.g. 10.1016/j.jcis.2019.10.020.
(3) The solvent effects generally include positive (e.g. 10.1002/adom.201800115) and negative (e.g. 10.1021/acsomega.0c05993) solvatochromism. The discussion of these two situations can strengthen the data analysis.
(4) It is really hard to understand the description of Figure 2 regarding the discussion of solvatochromism. I strongly recommend that the authors clearly describe the photoluminescence changes of 813-CDs with the increase/decrease of solvent polarity, at least including the red or blue shirt, increase or decrease in intensity (or QYs) of different emissive bands in the blue, green, and red regions. It will facilitate the discussion of the solvent effects to the band gaps of CDs.
(5) The authors have presented the formation of the resulting CDs according to the IR and DTA data. Could the authors also explain the emissive origins of their CDs?
(6) Actually, the PEI has been reported many times to modify CDs. For example, 10.1021/acssuschemeng.9b00325 and 10.3390/c7010002. Alternatively, the nitrogen content can also be increased by using ammonia, urea, or ethylenediamine. What is the reason that PEI is selected in this work?
(7) I cannot see the fluorescence well of 813-CD/PVP film if the authors only present a photograph in Figure 10A. So, please give photoluminescence spectra.
(8) The resolution of all figures has to be improved.
Reviewer 2 Report
The manuscript under review is focused on the synthesis and study of a new type of carbon dots with the property of photoluminescence in the red region of the spectrum. The authors synthesized red emissive carbon dots (813-CD). p-toluenesulfonate (IR-813) was used as a precursor. The manuscript presents a comprehensive study of various properties of the newly synthesized material. A very large amount of research has been done. The work can certainly be published in Molecules. However, in my opinion, the manuscript has some flaws regarding presentation.
My comments are given below. The manuscript can be published after these flaws are eliminated.
1. The introduction is written extremely inconsistently. For example, in the middle of the introduction, there is a sequence of sentences beginning with "In this study, we investigate the increased…." (line 40) and ending with the sentence "In summary, the optical Eg is extracted from the intercept…." (lines 51-52), which should be placed in section 2. Experimental Section. At the end of the introduction, the authors should also briefly outline the purpose of the work or list the most important results of this study.
2. Molecules is an open access journal and is published digitally. For this reason, in my opinion, the presentation of some of the results in the form of Supplementary information is not appropriate. It would be convenient for the reader to have all the information in one file. This could be achieved if all the figures included in Supplementary information were placed in the same article in the "Appendix" section.
3. In Fig. 1 on the abscissa Wavelength (nm) should be put instead of Wavelength (cm-1).
4. Caption to Fig. 2 is incomplete. It should be pointed out that the different colors of the lines in the PL spectra correspond to different wavelengths. The numeric values on the y-axis are unclear. In addition, they are expressed in an unknown unit “CPS”, with “CPS” being placed in brackets in some cases, and not in some other cases (same in Fig. 5).
5. In lines 181-183 the authors write that "Interestingly, 813-CDs were found to have a clear isosbestic point at 575 nm. This finding is important because it means that there will be no change in the optical property at a set wavelength. Hence, a uniformity in optical properties across the size distribution of CDs”. From the first three sentences, it is concluded that the optical properties do not depend on the particle size distribution. The following question arises: on what basis was such a conclusion made?
6. In my opinion, the results shown in Fig. 2 should be summarized in a form of a table, where the PL peaks for various solvents should be given in the order of their polarity decreasing.
7. The spectra of PL 813-CDs in DMSO and methylene chloride solvents at an excitation wavelength of 380 nm differ significantly from the PL spectra obtained for 813-CDs PL in other solvents. However, the authors do not discuss this detail in their article.
8. Figure 3 does not explain the double-peaked nature of 813-CDs PL in the studied solvents. In addition, in the same figure there are designations hvA, hvF and hv,F, which are not defined. In general, the caption to this figure is also incomplete.
9. In my opinion, Fig. 5 should be supplemented with a plot of the position of the second PL peak for 813-CD 25k PEI versus excitation wavelength. This would allow the reader to better follow the interesting pattern discovered by the authors.
10. Fig. 6 is not accurately presented. On the abscissa, instead of “Wavelength cm-1”, it should read “Wavelength (cm-1)”, and on the y-axis, numerical values ​​of the transmittance are plotted from 0 to 350%, which confuses the reader. In addition, the caption for Fig. 6 is not correct.
11. Fig. 8 should be rotated 90 degrees, otherwise it will be difficult for the reader to understand the presented data without printing it on paper.
12. The histograms presented in Fig. 9C and 9F cannot be approximated by a Gaussian distribution. Why do the authors try to approximate the presented histograms with a Gaussian distribution?
13. In general, the article requires proofreading. There are typos (for example, in line 50 n = 2 should be written instead of r = 2; in lines 96-97 "a resistivity of 18.2 M cm" should be "a resistivity of 18.2 MW cm", etc.), the authors arbitrarily handle commas and spaces (for example, a sentence starting in line 41 should end in line 46; however, the authors divided this sentence into two sentences, and, after formulas (1) and (2), there are no commas that are necessary; in the same sentence the optical pathlength symbol “l” should be written in italics). There is a strange symbol denoting degrees Celsius "40ºC " (see lines 111 and 125). Many similar remarks can be made. Authors should edit their article with more care.
Reviewer 3 Report
In this manuscript, the author reports, ‘Investigation into Red Emission and its Applications: Solvatochromic N-Doped Red Emissive Carbon Dots with Solvent Polarity Sensing and Solid-State Fluorescent Nanocomposite Thin Films”. The authors should address the following questions before getting a possible publication.
Recommendation: Major revisions are needed as noted.
1. The novelty of the present article should be discussed a little bit more in the Introduction section.
2. The formatting and grammatical errors in the article need to be checked carefully.
3. The author should write the purpose for each test in one/two sentences (in brief) before explaining the results of the characterization techniques.
4. The graphs in Fig.9 should be rearranged in a better way for improved representation.
5. The scale bar in Fig.10b is not clearly visible to the readers.
6. The authors are encouraged to include (1-2 sentences) future prospects of the present study in the conclusion.
7. A comparison table is required discussing about the fabrication methods, precursors, sizes, PL, and applications for the recently reported red emissive CDs.
8. The authors have cited relevant references in the Introduction section; however the manuscript needs to be highlighted with recent reports further to broaden the impact:
https://doi.org/10.1016/j.progsolidstchem.2020.100295 https://doi.org/10.3390/nano11082089
DOI 10.1149/2162-8777/abdfb8
https://doi.org/10.1021/acsapm.2c01579
Round 2
Reviewer 1 Report
The authors have solved my main comments.
The current manuscript is acceptable.
Reviewer 3 Report
The authors have addressed all the questions raised before therefore the manuscript can be accepted in its present form.